# Kinome Analysis to Define Mechanisms of Adjuvant Action: PCEP Induces Unique Signaling at the Injection Site and Lymph Nodes

**DOI:** 10.3390/vaccines10060927

**Published:** 2022-06-11

**Authors:** Sunita Awate, Erin Scruten, George Mutwiri, Scott Napper

**Affiliations:** 1UVAXX Pte. Ltd., 203 Henderson Industrial Road, Singapore 159546, Singapore; 2Vaccine and Infectious Disease Organization, 120 Veterinary Road, University of Saskatchewan, Saskatoon, SK S7N 5E3, Canada; erin.scruten@usask.ca (E.S.); george.mutwiri@usask.ca (G.M.); scott.napper@usask.ca (S.N.); 3School of Public Health, 107 Wiggins Road, University of Saskatchewan, Saskatoon, SK S7N 5E5, Canada; 4Department of Biochemistry, Microbiology, and Immunology, 107 Wiggins Road, University of Saskatchewan, Saskatoon, SK S7N 5E5, Canada

**Keywords:** kinome, adjuvant, PCEP, lymph nodes

## Abstract

Understanding the mechanism of action of adjuvants through systems biology enables rationale criteria for their selection, optimization, and application. As kinome analysis has proven valuable for defining responses to infectious agents and providing biomarkers of vaccine responsiveness, it is a logical candidate to define molecular responses to adjuvants. Signaling responses to the adjuvant poly[di(sodiumcarboxylatoethylphenoxy)phosphazene] (PCEP) were defined at the site of injection and draining lymph node at 24 h post-vaccination. Kinome analysis indicates that PCEP induces a proinflammatory environment at the injection site, including activation of interferon and IL-6 signaling events. This is supported by the elevated expression of proinflammatory genes (IFNγ, IL-6 and TNFα) and the recruitment of myeloid (neutrophils, macrophages, monocytes and dendritic cells) and lymphoid (CD4+, CD8+ and B) cells. Kinome analysis also indicates that PCEP’s mechanism of action is not limited to the injection site. Strong signaling responses to PCEP, but not alum, are observed at the draining lymph node where, in addition to proinflammatory signaling, PCEP activates responses associated with growth factor and erythropoietin stimulation. Coupled with the significant (*p* < 0.0001) recruitment of macrophages and dendritic cells to the lymph node by PCEP (but not alum) supports the systemic consequences of the adjuvant. Collectively, these results indicate that PCEP utilizes a complex, multi-faceted MOA and support the utility of kinome analysis to define cellular responses to adjuvants.

## 1. Introduction

By impacting the magnitude, duration, and nature, of the immune response, adjuvant selection can make, or break, a vaccine candidate [1]. Despite this, greater priority is often placed on antigens while relying on historic, empirical approaches to adjuvant selection and optimization. More recently, there has been priority to expand the panel of adjuvants available for human and livestock vaccines and to develop a greater understanding of their mechanisms of action (MOA) [2].

Many consequences of adjuvant administration have been described, ranging from localized depot effects [3], enhanced uptake and presentation of antigens [4,5], to higher-order impacts on immune response, including modulation of cytokine and chemokine profiles [6,7,8] and recruitment of immune cells [8,9]. Increasingly, there are efforts to elucidate different molecular MOA of adjuvants through omics approaches within the context of systems vaccinology [2]. Here the priority is to gain insight into the cellular responses induced by different adjuvants, at different biological locations, and to correlate these induced responses with various indicators of vaccine efficacy. To this objective, transcriptional responses has frequently been applied to define responses to common adjuvants [10,11,12,13,14]. From these efforts, a panel of “adjuvant core response genes” have been identified, as well as defining various molecular effectors including cytokines, chemokines, innate immune receptors, interferon-induced genes, and adhesion molecules [6]. While transcriptional profiling offers some advantages in terms of the maturity of the technology, ease of application, and magnitude of coverage, the findings must be interpreted with the caveat of potential disconnect between transcriptional and phenotypic responses due to post-transcriptional and post-translational regulatory events.

Kinase-mediated protein phosphorylation is the central mechanism for regulation of many cellular processes, including those associated with innate and adaptive immunity [15]. Investigations of global patterns of kinase-mediated signaling (kinome analysis) has emerged as a powerful tool to understand complex cellular biology [16]. Within that, kinome peptide arrays have demonstrated utility for deciphering immunity, including characterizing host-pathogen interactions [17,18,19], anticipating individual responses to vaccination [20], and describing cellular responses to discrete molecules (such as pathogen-associated molecular patterns) that are often employed as vaccine adjuvants [21,22]. Tissue and individual-specific signaling differences have been defined [23], supporting the use of the technology to define individualized responses to vaccines as well as characterizing different routes of vaccine delivery and unique sites of adjuvant action. The technology has proven robust in defining responses in biologically complex samples, including peripheral blood mononuclear cells [20], muscle and gut tissue [19,24], and whole insect homogenates [25]. Collectively, kinome analysis appears a strong candidate to be included within systems vaccinology approaches to define adjuvant MOA.

Polyphosphazenes are a novel category of adjuvants shown to be effective in a range of species through both parenteral and mucosal delivery [26,27]. These high molecular weight synthetic polymers consist of a backbone of alternating phosphorus and nitrogen atoms with organic side groups anchored to the phosphorus atoms [28,29]. Of the polphosphazenes, poly[di(sodiumcarboxylatoethylphenoxy)phosphazene] (PCEP), has demonstrated considerable ability to promote high-titre, long-lasting, immune responses to several antigens [30,31,32,33]. Through systemic and mucosal administration, including respiratory, oral, rectal, and intravaginal routes, PCEP promotes a balanced Th1/Th2 type response [26,27], and functions well in combination with other adjuvants [34]. These characteristics prompted interest to advance PCEP as an adjuvant for human and veterinary applications.

There is growing appreciation of the biological complexity of the MOA of polphosphazenes. Indications of their immunostimulatory properties indicate that the formation of depots at the injection site offer minimal contribution to their adjuvant activity [31], although complexes with the antigen do facilitate antigen delivery to immune cells [35]. Recent efforts attribute their activity to stimulation of immune responses at the injection site [4], splenocytes [27,36], and lymph nodes [37]. The potential of PCEP as an adjuvant for human and animal vaccines, coupled with indications of a diverse and biological complexity MOA, make it an ideal candidate for investigation of kinome analysis to describe molecular responses to adjuvants.

## 2. Materials and Methods

### 2.1. Adjuvants

PCEP was generated by Idaho National Laboratory (Idaho Falls, ID, USA) using described methods [36,37]. Endotoxin levels within PCEP were determined to be less than 0.034 ng/mL as assessed by the Limulus Amebocyte Lysate assay (Biowhittaker, Walkersville, MD, USA). PCEP was dissolved in Dulbecco’s phosphate buffered saline (PBS) (Gibco, Grand Island, NY, USA). The alum adjuvant (Thermo Fisher Scientific, Rockford, IL, USA) is a mixture of alum and magnesium hydroxide (40 mg/mL).

### 2.2. Mouse Trials

Female BALB/c mice (purchased from Charles River Laboratories, North Franklin, CT, USA) of 4–6 weeks of age were used in these experiments. The animal experiments were approved by the University of Saskatchewan’s Animal Research Ethics Board and adhered to the Canadian Council on Animal Care guidelines for humane use of animals.

### 2.3. Quantitative Real-Time PCR (qRT-PCR)

Female BALB/c mice (*n* = 6) were injected intramuscularly (i.m) in the quadriceps muscle with 25 μL of either phosphate-buffered saline (PBS) as control or 50 μg PCEP per animal. As the trauma caused by injecting a liquid into the tissue is sufficient to alter gene expression [6,38], the PBS-injected group is an important control. Mice were euthanized and samples collected at 24 h post-injection.

Immediately after mice were euthanized, whole muscle tissues from the thigh were collected in TRIzol (Invitrogen) and aseptically homogenized with 2.3 mm Zirconia microbeads (Biospec Products Inc., Bartlesville, OK, USA) in a Mini-Beadbeater^TM^ (Biospec Products Inc., Bartlesville, OK, USA). The homogenates were centrifuged for 1 min at 10,000× *g*, and the supernatants were collected for total RNA extraction as per the manufacturer’s instruction. The extracted RNA was quantified and treated with DNase (Invitrogen). The cDNA was synthesized using random hexamers (Applied Biosystems) and SuperScript^®®^ II Reverse Transcriptase (Invitrogen) as per manufacturer’s instruction. All PCR reactions were carried out in duplicate in 96-well plates with optical quality tape (Bio-Rad) using an iCycler iQ^®®^ Real-Time PCR Detection System (Bio-Rad, Hercules, CA, USA). Each PCR reaction contained 1 μL target cDNA, 0.2 μM each of forward and reverse primers, 7.5 μL of iQ SYBR^®®^ Green Supermix (Invitrogen) and distilled water to 15 μL of final volume according to manufacturer’s instruction. The negative control contained all the reagents except cDNA. All the primers used in quantitative RT-PCR are shown in Table 1. Reference genes GAPDH, RPL19 and18 s rRNA were analyzed and the best (GAPDH) was selected for further analysis. Amplification was performed by initial denaturation at 95 °C for 3 min in cycle 1, followed by cycle 2 (95 °C, 15 s; 55 °C, 30 s; 72 °C, 30 s) ×45 and then cycle 3, the Melt curve analysis, was pre-set at 55 °C ramping to 95 °C with 1 °C increase each 10 s and final hold at 20 °C. A Melt Curve analysis was performed to ensure that any product detected was specific to the desired amplicon.

### 2.4. Isolation of Recruited Cells from Site of Injection and Draining Lymph Node

Female BALB/c mice were divided into three groups (*n* = 5 per group) and were injected i.m on both legs with 25 μL of either PBS as control, 50 μg PCEP, or 0.5 mg of alum. At 24 h post-injection, muscle tissues were dissected from the site of injection, minced, and incubated with digestion buffer (Hank’s Balanced Salt Solution [HBSS] [Gibco] supplemented with 0.1% type II collagenase D [Worthington Biochemical, NJ, USA], 0.2% BSA [Sigma–Aldrich, MO, USA], 0.025% trypsin [Gibco] and 0.01% DNase I [Roche Diagnostics, Germany]) for 45 min at 37 °C under constant agitation. The cell suspension was filtered through 70 μm cell strainer and layered on 25% percoll (GE healthcare, Chicago, IL, USA) and centrifuged at 2000× *g* for 1 h. The cell pellets were washed twice, resuspended in RPMI (Gibco) with 10% FBS (Gibco), and used for fluorescent labeling for FACS analysis. Cell viability was estimated by Trypan Blue (Gibco) exclusion.

The draining inguinal lymph nodes were dissected, minced, and incubated with digestion buffer containing 2 mg/mL collagenase D (Roche Diagnostics, Mannheim, Germany) and 0.25 mg/mL DNase I in HEPES (Gibco) for 15 min at 37 °C. These samples were then filtered through 70 μm cell strainer to obtain a single cell suspension which was used for fluorescent labeling for FACS analysis.

### 2.5. Flow Cytometry

For FACS analysis, all the cells were stained in the presence of an Fc block. For staining, cells were incubated for 20 min at 4 °C using the following antibodies: CD11b-FITC, Ly6C-APC, Ly6G-APC, F4/80- PE, CD11c-PE, CD3-APC, CD8-FITC, CD4-FITC, CD19-FITC (all from eBiosciences, CA, USA) and CD8-PerCP-Cy5.5, CD4-PerCP-Cy5.5 (all from BD Biosciences). The following markers were used to identify specific cell types: monocytes (CD11b+ Ly6C+), neutrophils (CD11b+ Ly6G+), macrophages (F4/80), dendritic cells (CD11c), B cells (CD19), CD4+ T cells (CD3+ CD4+) and CD8+ T cells (CD3+ CD8+).

Unstained cells were used to set up the instrument. Compensation controls were set up using single stains and isotype controls were used to determine the level of non-specific binding. The cells were gated based on simple forward and side scatter patterns. Furthermore, all the dead cells were excluded using the viability dye, propidium iodide and doublet discrimination was performed by plotting FSC-H vs. FSC-A. If we take the example of lymphocytes, they were identified and gated by their forward and side scatter patterns. The CD3+ T cells were then further identified and gated by the expression of CD4+ and CD8+. The expression of surface markers was assessed using CellQuest analysis software on a FACSCalibure flow cytometer (BD Biosciences).

### 2.6. Peptide Arrays for Kinome Analysis

Female BALB/c mice were divided into three groups (*n* = 5) and were injected i.m on both legs with 25 μL of either PBS as control, 50 μg PCEP, or 0.5 mg of alum. At 24 h post injection, muscle tissues were dissected from the site of injection, minced, and incubated with digestion buffer (Hank’s Balanced Salt Solution [HBSS] [Gibco] supplemented with 0.1% type II collagenase D [Worthington Biochemical, Lakewood, NJ, USA], 0.2% BSA [Sigma–Aldrich, MO, USA], 0.025% trypsin [Gibco] and 0.01% DNase I [Roche Diagnostics, Germany]) for 45 min at 37 °C under constant agitation. The cell suspension was filtered through 70 μm cell strainer and layered on 25% percoll (GE healthcare, Chicago, IL, USA) and centrifuged at 2000× *g* for 1 h. The cell pellets were washed thrice with ice-cold PBS and stored at −20 °C until further use.

Similarly, the draining inguinal lymph nodes were dissected, collected, minced, and incubated with digestion buffer containing 2 mg/mL collagenase D (Roche Diagnostics, Mannheim, Germany) and 0.25 mg/mL DNase I in HEPES (Gibco) for 15 min at 37 °C. It was then filtered through 70 μm cell strainer to obtain a single cell suspension. The cells were washed thrice with ice-cold PBS and stored at −20 °C until further use.

Protocols for the design and application of the peptide arrays have been described [20]. Arrays were constructed by a commercial provider (JPT Innovative Peptide Solutions, Berlin, Germany and designed to include peptides representing phosphorylation events associated with a wide variety of signaling pathways. Each array includes nine technical replicates of each of the 282 unique peptides. All kinome experiments were performed on the same day to minimize potential inter-assay variance.

### 2.7. Analysis of Kinome Data

Peptide-spot intensities were transformed using a variance-stabilizing normalization (VSN) method through the online software, PIIKA 2.0 (https://saphire.usask.ca/saphire/piika2.0/) [39]. Peptides that showed variation in technical replicates via Chi-squared test (χ^2^ < 0.01) were removed from subsequent analysis. Consistent technical replicates were averaged together, and fold-change (FC) for each peptide was calculated as previously described [20]. Theta-distributed stochastic neighbour embedding(t-SNE) analysis and hierarchical clustering were conducted using peptides with consistent phosphorylation (χ^2^ > 0.01). The t-SNE analysis was conducted using the R package Rtsne (https://github.com/jkrijthe/Rtsne) (accessed on 14 June 2021) and visualized using ggplot2 (https://ggplot2.tidyverse.org) (accessed on 14 June 2021). The t-SNE analysis was performed 100 times and the result with the lowest value of the objective function was selected. The construction of the heatmap using PIIKA 2.0 has been previously [39]. Hierarchical clustering was conducted using the Pearson correlation distance and McQuitty linkage. Peptides were considered differentially phosphorylated under two given criteria: first, the peptide was consistently phosphorylated according to the Chi-squared test and second, the VSN-transformed phosphorylation intensity of an individual peptide was significantly different (two-tailed Welch’s t-test for Unequal Variances, *p* < 0.05) between cohorts.

### 2.8. Pathway Over-Representation Analysis

Peptides that were differentially phosphorylated were subjected to pathway over-representation analysis (ORA) using InnateDB [40]. ORA was completed using the hypergeometric algorithm with Benjamani–Hochberg correction method, and pathways were considered statistically significant with a false discovery rate (FDR) of *p* < 0.05.

### 2.9. Statistical Analysis

The increase in target gene expression levels in PCEP stimulated muscle tissues were calculated as fold change increase (2^−ΔΔCT^). The data for cell recruitment were analyzed using Graph-Pad Prism 6 software (GraphPad Software, San Diego, CA, USA). Differences in the cell numbers between the treatments were analyzed by two-way ANOVA by Ranks and the significant differences between the treatments were compared by Bonferroni multiple-comparison test where **** *p* < 0.0001, *** *p* < 0.001, ** *p* < 0.005, * *p* < 0.05.

## 3. Results

### 3.1. Kinome Analysis

Hierarchal clustering of the kinome datasets indicates tissue-specific differences in signaling between the injection site and draining lymph nodes (Figure 1A). These differences are anticipated given the distinct functions of these tissues [41]. Hierarchal clustering also offers indication of the relative magnitude of the differential responses to each adjuvant at the lymph node; close clustering of the alum and PBS datasets, relative to PCEP, indicates the more dramatic consequences of PCEP. These differential magnitudes of response are quantified through consideration of the number of peptides which are differentially phosphorylated peptide in response to each adjuvant; seven for alum and 98 for PCEP (Figure 1B). In comparing the tissue-specific responses, PCEP induces pronounced signaling responses at both the injection site (98 differentially phosphorylated peptides) and the lymph nodes (86 differentially phosphorylated peptides), but these responses are quite distinct, indicating PCEP induces systemic, tissue-specific responses (Figure 1C).

### 3.2. Signaling Events at the Injection Site

At the site of injection, administration of PBS is likely to cause localized tissue damage that result in cellular responses [6,38]. By comparing the responses to adjuvant relative to the PBS control of the same tissue enables identification of adjuvant-specific signaling responses. At the site of injection, 87 peptides show significant (*p* < 0.05) differences in phosphorylation levels in response to PCEP relative to the PBS control (Table 2). Within these differentially phosphorylated peptides, there is an approximately equal proportion of peptides with increased or decreased phosphorylation. Pathway overrepresentation analysis of these proteins associated with these phosphorylation events indicates activation of pro-inflammatory immune responses. Specifically, the patterns of peptide phosphorylation indicate activation of interferon-mediated signaling as well as activation of branches of the innate immune response, including Toll-like receptor and interleukin signaling (Table 3).

### 3.3. Validation of Signaling Events at Site of Injection

The proinflammatory responses to PCEP were investigated through quantitative RT-PCR of a panel of known pro-inflammatory genes. Consistent with the kinome data, there is increased expression of pro-inflammatory genes, most notably for Il-6 (Figure 2B) but also for TLRs, TNFα and IFNγ (Figure 2A). Increased expression of members a family of chemokine receptors, which is also characteristic of proinflammatory responses, is observed at the injection site (Figure 2C). The functional consequences of PCEP, and further evidence of the pro-inflammatory responses induced by this adjuvant, are supported by patterns of cell migration to the injection site. A variety of immune cells, including myeloid cells (neutrophils, macrophages, monocytes and dendritic cells) (Figure 3A) and lymphoid cells (CD4+, CD8+, and B cells), were recruited to the injection site (Figure 3B).

### 3.4. Signaling Events at the Draining Lymph Nodes

At the draining lymph nodes, 98 peptides show significant (*p* < 0.05) differences in phosphorylation levels in response to PCEP at 24 h post-injection (Table 4). By comparison, alum induced differential phosphorylation of only 7 peptides (Table 5). This indicates that the impact of alum is largely localized to site of injection, whereas PCEP induces more systemic immune responses. Pathway overrepresentation analysis indicates that PCEP induces pro-inflammatory innate immune responses at the draining lymph node, as indicated by the Jak-Stat signaling pathway, interleukin signaling pathways (including Il-2, Il-6, and Il-7), as well as activation of pathways associated with vascular endothelial growth factor (VEGF), erythropoietin (EPO) and transforming growth factor beta (TGFβ) (Table 6). The small number of differential phosphorylated peptides at the lymph nodes in response to alum negates the ability to perform pathway analysis.

### 3.5. Validation of Signaling Events at Draining Lymph Nodes

The differential impacts of alum and PCEP on signaling at the lymph node are supported by patterns of immune cell migration; in response to PCEP all of categories of immune cells considered, including macrophages, neutrophils, monocytes, dendritic cells, B cells, CD4+, and CD8+ cells, were significantly higher in lymph nodes of animals administered with PCEP (Figure 4). In contrast, fewer categories of immune cells, and to lesser degrees, were impacted by the administration of alum.

## 4. Discussion

Within this investigation, kinome profiling was performed to define signaling events to PCEP at the injection site and draining lymph node at twenty-four hours post-injection. At the injection site, PCEP induced pro-inflammatory signaling, as exemplified by the Jak-Stat pathway. PCEP-induced activation of interferon-based signaling was supported by elevated expression of proinflammatory genes as well as recruitment of immune cells. This supports the hypothesis that an element of the MOA of PCEP resides in its ability to promote a pro-inflammatory environment at the injection site. Kinome analysis also indicates that responses to PCEP are not restricted to the injection site as strong signaling responses, relative to alum and distinct from those observed at the injection site, occur at the draining lymph node. Consistent with that, PCEP resulted in significantly higher recruitment of immune cells to the lymph node than alum. Collectively, these results indicate that the complex, multi-faceted adjuvant activity of PCEP and support the utility of kinome analysis to define adjuvant MOA.

In 2015, Hagan and Fox predicted a “New Golden Age” for vaccine adjuvants [42]. This enthusiastic assessment was based on an expanding knowledge of adjuvant MOA that was largely enabled through systems vaccinology approaches. While initially overlooked, we now have fuller appreciation of the ability of adjuvants to improve the range, practicality, and efficacy of vaccines through dose sparing, enabling rapid immune responses, broadening of the induced antibody response, and optimizing of the magnitude of the vaccine-associated antibody response [1,2]. Adjuvants can also be the critical determinants of new categories of vaccines, including for the induction of T cell responses, mucosal vaccines, and personalized vaccines. This includes interest in individual adjuvants with these characteristics as different formulations and co-formulations. Within this power to impact immune responses, there is also the appreciation of the potential for unintended, and potentially detrimental responses, both at the site of administration as well as systemic consequences [43].

The value of systems vaccinology is to identify molecular responses to immunization to identify surrogate markers of immunogenicity and reactogenicity that anticipate whether the patient will develop the desired immune response (correlates of immunity) and/or will be protected from the targeted disease (correlates of protection), as well as to understanding the underlying mechanisms of these outcomes [44]. These responses occur at the site of injection as well as other immune related locations, such as peripheral blood mononuclear cells and lymph nodes. While systems vaccinology approaches have largely been grounded in transcriptional analysis, that various adjuvants are known to activate signaling pathways associated with innate and adaptative immune responses, coupled with recent advances in technologies for defining global patterns of phosphorylation-mediated signal transduction, there is both opportunity and priority to apply kinome analysis to define adjuvant MOA. The motivations for the current investigation were to define biological responses to an important adjuvant, including considerations of localized and systemic effects, as well as to investigate kinome analysis as a tool to define adjuvant MOA.

With respect to the MOA of PCEP, the current work supports the hypothesis this adjuvant functions through both localized and systemic immune responses. At the site of injection, there is clear indication of activation of proinflammatory signaling beyond that resulting from general tissue damage [38]. This offers mechanistic explanation and additional dimension of the observed patterns of induced expression of a variety of proinflammatory genes. There have also been indications that the MOA of PCEP is not limited to the site of injection with indication of activation of higher-order immune response as suggested by patterns of increased patterns migration of immune cells to lymph nodes in response to PCEP administration [37]. With that, however, there was minimal information about the biochemical basis of these changes, including potential insights into biomarkers, as well as comparable analysis of these changes to other adjuvants. The kinome analysis of the responses within the lymph nodes to PBS, alum, and PCEP provides context of the magnitude of systemic responses induced by PCEP relative to alum; a difference of 100 versus 7 differentially phosphorylated peptides. Within those signaling responses, there are the anticipated changes to signaling associated with inflammatory responses as well as activation of erythropoietin (EPO) mediated signaling. Within mouse models, it has been demonstrated that administration, or engineered overexpression, of EPO increased humoral antibody responses to several antigens [45]. Similarly, administration of EPO to patients with chronic kidney disease improved vaccine responsiveness [46]. Erythropoietin treatment is also associated with an augmented immune response to the influenza vaccine in hematologic patients [47]. Collectively this suggests a role for EPO in humoral immune responses. The implication that PCEP administration influences EPO signaling within the lymph nodes merits further investigation as a potential MOA. Collectively, the results of this investigation highlight and detail the complex, multi-faceted MOA of PCEP while supporting the utility of kinome analysis as a tool to define responses to adjuvants.

## Figures and Tables

**Figure 1 vaccines-10-00927-f001:**
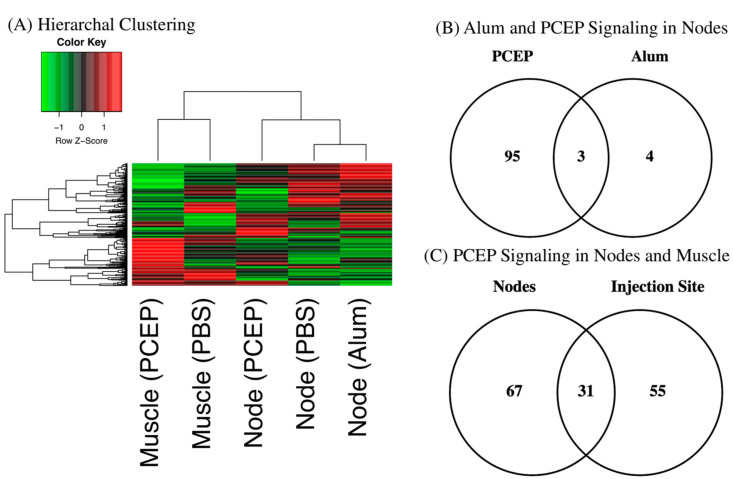
Kinome Responses to PCEP and Alum at Site of Injection and Lymph Node. (**A**) Clustering of Kinome Responses at Site of Injection and in Draining Lymph Nodes. Hierarchical clustering of kinome datasets. (1−Pearson correlation) was used as the distance metric, while McQuitty linkage was used as the linkage method. Colors indicate the average (over nine intra-array replicates) normalized phosphorylation intensity of each target, with red indicating greater amounts of phosphorylation and green indicating lesser amounts of phosphorylation. (**B**) Venn Diagram of Differentially Phosphorylated Peptides in Lymph Nodes to PCEP and Alum. Comparison of peptides consistently and significantly (*p* < 0.05) phosphorylated in lymph in response to either PCEP or Alum relative to PBS control. (**C**) Venn Diagram of Differentially Phosphorylated Peptides in Lymph Nodes and Muscle to PCEP. Comparison of peptides consistently and significantly (*p* < 0.05) phosphorylated in lymph and muscle in response to PCEP relative to PBS control.

**Figure 2 vaccines-10-00927-f002:**
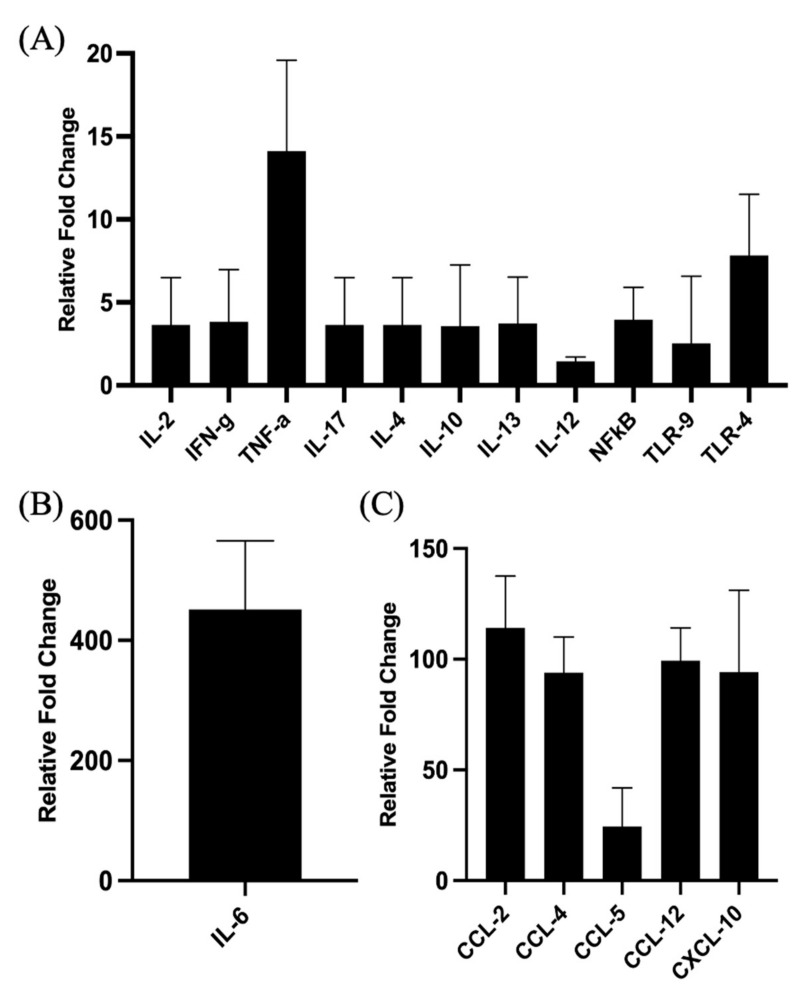
Patterns of Gene Expression at Site of PCEP Injection. Cytokine and chemokine gene expression profiles elicited by PCEP at the site of injection after intramuscular injection in mice. Mice were injected with PBS or PCEP intramuscularly. Muscle tissue at the site of injection were collected at 24 h and analyzed for cytokine and chemokine genes by quantitative real-time PCR. (**A**) Increased expression of cytokine genes including IFNγ, TNFα and TLRs at the injection site. (**B**) Substantial increase in proinflammatory gene, IL-6. (**C**) Increased expression of chemokine receptor family genes at the injection site. Results shown are the mean ± SE of six replicates at each time point. Relative fold changes (*y*-axis) for each gene were normalized to mouse GAPDH. Fold changes are calculated by the Ct method and are relative to the gene expression in PBS injected muscle tissue. *(reprinted with modifications from Awate et al. 2012, Copyright 2012, with permission from Elsevier).*

**Figure 3 vaccines-10-00927-f003:**
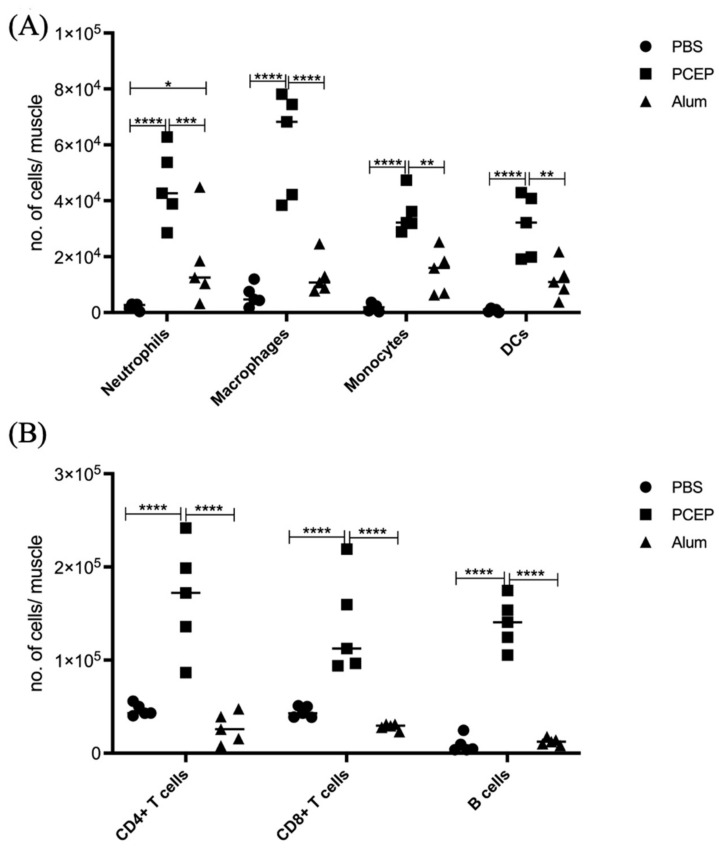
Patterns of Cell Recruitment to the Site of Injection in Response to Adjuvants. PCEP stimulates increased immune cell numbers at the site of injection. BALB/c mice (*n* = 5 per group) were injected i.m. with either PBS, PCEP (50 ug) or alum (0.5 mg). The site of injection muscle tissue was dissected at 24 h time point and processed. Single cell suspensions were analyzed by flow cytometry. (**A**) Kinetics of myeloid cells (neutrophils, macrophages, monocytes and dendritic cells) 24 h post-injection of adjuvants at the site of injection. (**B**) Kinetics of lymphoid cells (CD4+, CD8+ and B cells) 24 h post-injection of adjuvants at the site of injection. Differences in the cell numbers were analyzed by two-way ANOVA and the significant differences between the treatments were compared by Bonferroni multiple-comparison test where **** *p* < 0.0001, *** *p* < 0.001, ** *p* < 0.005, * *p* < 0.05. *(reprinted with modifications from Awate et al. 2014, Copyright 2014, with permission from Elsevier).*

**Figure 4 vaccines-10-00927-f004:**
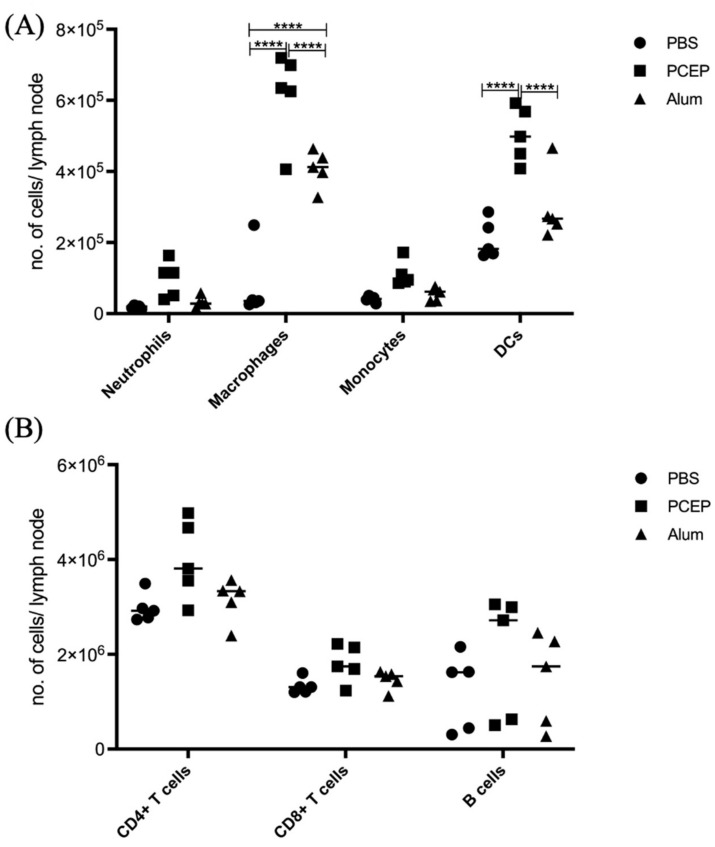
Patterns of Cell Recruitment to the Draining Lymph Nodes in Response to Adjuvants. PCEP stimulates increased immune cell numbers in the draining lymph nodes. BALB/c mice (*n* = 5 per group) were injected i.m. with either PBS, PCEP (50 ug) or alum (0.5 mg). The draining inguinal lymph nodes were collected at 24 h post-injection and the cell suspensions were analyzed by flow cytometry. (**A**) Kinetics of myeloid cells (neutrophils, macrophages, monocytes and dendritic cells) 24 h post-injection of adjuvants at the draining lymph nodes. (**B**) Kinetics of lymphoid cells (CD4+, CD8+ and B cells) 24 h post-injection of adjuvants at the draining lymph nodes. Differences in the cell numbers were analyzed by two-way ANOVA and the significant differences between the treatments were compared by Bonferroni multiple-comparison test where **** *p* < 0.0001. *(reprinted with modifications from Awate et al. 2014, Copyright 2014, with permission from Elsevier).*

**Table 1 vaccines-10-00927-t001:** Primer Sequences.

S. No	Gene Symbol	Forward Primer	Reverse Primer
1	IL-2	CCTGGAGCAGCTGTTGATGG	CAGAACATGCCGCAGAGGTC
2	IL-4	ATGGGTCTCAACCCCCAGC	GCTCTTTAGGCTTTCCAGG
3	IL-6	TGTCTATACCACTTCACAAGTC	GCACAACTCTTTTCTCATTTCCA
4	IL-10	TAGTTCCCAGAAGCCATGTG	AGAGGGAGCAGTTTGTAAGC
5	IL-12	TGCCAGCCTGCCTTATATTG	TCCACCAGGACCACTAAATG
6	IL-13	CAGCAGCTTGAGCACATTTC	CATAGGCAGCAAACCATGTC
7	IL-17	ACCTCAACCGTTCCACGTCA	CAGGGTCTTCATTGCGGTG
8	IFN-γ	TGAACGCTACACACTGCAT	CGACTCCTTTTCCGCTTCCT
9	TNF-α	GACCCTCACACTCAGATCATCT	CCACTTGGTGGTTTGCTACGA
10	NFκB	AGAAGACACGAGGCTACAAC	TCACAGACGCTGTCACTATC
11	TLR-4	TCCCAGTGATGGCTGATTAG	GCACCCAACATTGTGTTACC
12	TLR-9	GAAGGGACAGCAATGGAAAG	GCCAAGTGCTACCATTAACC
13	CCL-2	TCACCTGCTGCTACTCATTC	TCTGGACCCATTCCTTCTTG
14	CCL-4	CCAGCTGTGGTATTCCTGAC	GAGCTGCTCAGTTCAACTCC
15	CCL-5	CTCCCTGCTGCTTTGCCTAC	CACACTTGGCGGTTCCTTCG
16	CCL-12	TGCCTCCTGCTCATAGCTAC	GGCTGCTTGTGATTCTCCTG
17	CXCL-10	GTCACATCAGCTGCTACTCC	CGCACCTCCACATAGCTTAC

**Table 2 vaccines-10-00927-t002:** Differentially Phosphorylated Peptides in Muscle is Response to PCEP.

Increased Phosphorylation					Decreased Phosphorylation				
**ID**	**P Site**	**Accession**	**FC**	** *p* **	**ID**	**P Site**	**Accession**	**FC**	** *p* **
Shc1	Y439	P29353	1.59	0.02	Lyn	Y396	P07948	−1.5	0.05
PLCG2	Y759	P16885	1.49	0.05	IRF-3	S402	Q14653	−1.37	0.04
Smad3	S423	P84022	1.47	0.03	IKK-g	S43	Q9Y6K9	−1.35	0.01
Syk	Y352	P43405	1.44	0.01	IKK-b	Y188	O14920	−1.3	0.03
Syk	Y525	P43405	1.43	0.01	MDM2	S166	Q00987	−1.3	0.004
Ck2-B	S228	Q5SRQ6	1.35	0.04	ACTA1	Y55	P68133	−1.29	0.02
Cdk4	S150	P11802	1.34	0.02	MAVS	S233	Q7Z434	−1.28	0.04
p300	S2279	Q09472	1.34	0.05	Keap1	S293	Q14145	−1.28	0.02
EP300	S2366	Q09472	1.33	0.0001	MK2	Y132	P16389	−1.27	0.05
CTNNB1	Y654	P35222	1.31	0.005	IKK-a	S180	O15111	−1.26	0.0002
TAK1	T178	O43318	1.3	0.02	p38-a	Y322	Q16539	−1.25	0.04
SEK1	T261	P45985	1.3	0.04	IRAK1	T100	P51617	−1.25	0.02
TAK1	T187	O43318	1.29	0.04	IL7R	Y449	P16871	−1.25	0.004
Grb10	S150	Q13322	1.28	0.01	PDK1	S241	O15530	−1.24	0.001
K8	S74	P05787	1.27	0.02	Fos	S362	P01100	−1.23	0.05
Sek1	S80	P45985	1.27	0.03	MEK1	Y385	Q02750	−1.23	0.03
EGFR	T693	P00533	1.26	0.007	CREB	S133	P16220	−1.23	0.02
SOC3	Y221	O14543	1.26	0.02	Jun	S63	P05412	−1.22	0.05
TBK1	S172	Q9UHD2	1.25	0.02	HSP70	Y525	P08107	−1.22	0.04
Smad6	S435	O43541	1.24	0.02	Met	Y1003	P08581	−1.22	0.01
Cdc42	Y32	P60953	1.24	0.03	ACC1	S29	Q13085	−1.22	0.004
XIAP	S87	P98170	1.23	0.02	Akt1	T308	P31749	−1.21	0.05
IRAK4	T208	P51617	1.21	0.01	JNK2	T183	P45984	−1.2	0.03
SMAD3	S204	Q15796	1.21	0.03	PDK1	Y373	O15530	−1.2	0.01
STMN1	S24	P16949	1.21	0.05	Casp3	S150	P42574	−1.19	0.05
CDK2	Y14	P24941	1.2	0.006	Mapk14	T122	Q16539	−1.19	0.03
TrKA	Y496	P04629	1.2	0.02	Aura	T287	O14965	−1.18	0.04
P27kip1	Y74	P46527	1.2	0.03	JNK1	T183	P45983	−1.18	0.02
Crk	Y221	P46108	1.19	0.02	Bim	S69	O43521	−1.17	0.02
IRAK1	T387	P51617	1.18	0.01	p70S6K	S447	P23443	−1.16	0.005
TNIK	T181	Q9UKE5	1.18	0.02	Mnk1	T255	Q9BUB5	−1.15	0.05
Tyk2	Y1054	P29597	1.18	0.03	CHOP	S79	P35638	−1.15	0.05
EGFR	Y869	P00533	1.18	0.05	Mek1	S217	Q02750	−1.15	0.02
TrKA	Y757	P04629	1.17	0.008	CDK2	T160	P24941	−1.15	0.02
gp130	Y767	P40189	1.17	0.02	MSK2	S360	O75676	−1.14	0.02
CREB	S117	P16220	1.17	0.02	NFkB p65	S536	Q04206	−1.14	0.01
SHC3	Y341	Q92529	1.17	0.03	CREB	S111	P16220	−1.13	0.05
PI3K p85	Y605	O00459	1.17	0.05	Fyn	Y420	P06241	−1.13	0.02
Smad3	T179	P84022	1.17	0.05	Met	Y1234	P08581	−1.11	0.02
IFNAR1	Y466	P17181	1.15	0.03					
STAT6	Y641	P42226	1.14	0.03					
HSP60	S70	P10809	1.14	0.03					
GIT2	Y592	Q14161	1.13	0.01					
STMN1	S37	P16949	1.12	0.03					
IFNGR1	S495	P15260	1.11	0.02					
SOC3	Y204	O14543	1.09	0.003					
CTNNB1	S33	P35222	1.07	0.02					

**Table 3 vaccines-10-00927-t003:** Pathway Over-Representation Analysis PCEP Site of Injection.

Pathway Name	Pathway ID	Source Name	Pathway Uploaded	Pathway *p*-Value
JAK STAT pathway and regulation	16125	INOH	27	5.04 × 10^23^
RANKL	15925	NETPATH	19	1.20 × 10^22^
Fc epsilon receptor signaling	17802	REACTOME	23	2.07 × 10^22^
Innate Immune System	17476	REACTOME	33	3.51 × 10^22^
IL-7 signaling	16106	INOH	23	3.54 × 10^22^
Pathways in cancer	4397	KEGG	27	1.98 × 10^21^
EPO signaling pathway	16151	INOH	22	8.75 × 10^21^
VEGF signaling pathway	16190	INOH	22	1.01 × 10^20^
Immune System	18444	REACTOME	40	2.63 × 10^20^
BCR signaling pathway	15384	PID NCI	16	3.06 × 10^20^
IL2	15918	NETPATH	17	4.59 × 10^20^
Toll-like receptor signaling pathway	564	KEGG	18	8.14 × 10^20^

**Table 4 vaccines-10-00927-t004:** Differentially Phosphorylated Peptides in Node in Response to PCEP.

Increased Phosphorylation					Decreased Phosphorylation				
ID	P Site	Accession	FC	*p*	ID	P Site	Accession	FC	*p*
p47phox	S370	P14598	1.71	0.003	PKACa	S10	P17612	−1.74	0.03
NFAT3	S676	Q14934	1.69	0.009	P300	S89	Q09472	−1.61	0.007
P27kip1	T157	P46527	1.64	0.004	PKACa	S338	P17612	−1.54	0.04
SHC3	Y341	Q92529	1.62	0.004	IKK-beta	Y188	O14920	−1.54	0.03
K8	S74	P05787	1.58	0.02	Lyn	Y396	P07948	−1.53	0.01
STAT1	S708	P42224	1.57	0.006	p70S6K	S447	P23443	−1.5	0.02
Mek2	S226	P36507	1.57	0.04	MyD88	Y257	Q99836	−1.49	0.04
IKK-alpha	S473	O15111	1.55	0.02	Jak2	Y813	O60674	−1.48	0.02
Rack1	Y194	P63244	1.52	0.0004	PKACa	T197	P17612	−1.46	0.005
CHOP	S79	P35638	1.51	0.02	p67phox	S208	P19878	−1.46	0.0008
STAT1	S727	P42224	1.49	0.02	MSK2	S360	O75676	−1.43	0.01
MK2	Y415	P16389	1.48	0.01	IKK-a	S180	O15111	−1.42	0.02
Rab5A	S123	P20339	1.47	0.01	Lyn	Y507	P07948	−1.42	0.02
DVL1	S679	O14640	1.46	0.0002	PPARG	S112	P37231	−1.41	0.04
PI3K p85 b	Y464	O00459	1.46	0.05	Jak2	Y119	O60674	−1.41	0.04
ACC1	S80	Q13085	1.45	0.02	Mnk1	T250	Q9BUB5	−1.41	0.03
Smad6	S435	O43541	1.41	0.02	p67phox	T233	P19878	−1.41	0.01
EP300	S2366	Q09472	1.4	0.007	MAPK14	T179	Q16539	−1.41	0.007
4E-BP1	T46	Q13541	1.39	0.04	PPP2CA	T304	P67775	−1.4	0.05
Flt3	Y842	P36888	1.37	0.05	Pyk2	S213	Q14289	−1.38	0.01
Rab5A	Y205	P20339	1.36	0.01	Mnk1	T255	Q9BUB5	−1.37	0.008
IRAK4	T208	P51617	1.34	0.01	NFAT1	S326	Q13469	−1.36	0.01
Crk	Y221	P46108	1.34	0.04	STMN1	S15	P16949	−1.34	0.02
IFNGR1	S495	P15260	1.33	0.006	NFAT1	S110	Q13469	−1.32	0.01
Grb10	S150	Q13322	1.33	0.02	IL4R	Y713	P24394	−1.32	0.01
Cdc42	Y64	P60953	1.32	0.02	MEK1	T385	Q02750	−1.31	0.04
Met	Y1003	P08581	1.32	0.02	STAT5B	S731	P51692	−1.3	0.01
TGFBR1	T204	P36897	1.31	0.03	IKK-g	S43	Q9Y6K9	−1.29	0.03
Sek1	S80	P45985	1.3	0.02	SMAD3	S416	Q15796	−1.28	0.01
CTNNB1	Y654	P35222	1.3	0.05	Tgfbr2	S409	P37173	−1.27	0.005
Calmodulin	Y99	P62158	1.27	0.03	Keap1	S293	Q14145	−1.26	0.03
Cdc2	T161	P06493	1.27	0.05	STAT4	S722	Q14765	−1.25	0.05
BRAF1	S579	P15056	1.26	0.02	Jun	S63	P05412	−1.25	0.04
IKK-alpha	T23	O15111	1.25	0.02	PIK3R1	Y528	P27986	−1.2	0.05
Cdk4	S150	P11802	1.25	0.02	PDGFRb	Y686	P09619	−1.2	0.004
NFkB-p65	S276	Q04206	1.24	0.01	PDK1	Y376	O15530	−1.19	0.05
Rab4	Y189	P20338	1.24	0.03	Pyk2	S399	Q14289	−1.19	0.05
Rack1	Y52	P63244	1.21	0.03	PDGFRb	Y740	P09619	−1.17	0.004
TBK1	S172	Q9UHD2	1.2	0.02	Jak1	Y220	P23458	−1.16	0.04
TrKA	Y680	P04629	1.19	0.04	PDK1	S241	O15530	−1.15	0.02
Grb2	Y37	P62993	1.18	0.02	MAPK14	T122	Q16539	−1.14	0.04
caveolin-1	Y6	Q03135	1.18	0.03	TAB1	S423	Q15750	−1.12	0.05
Shc1	Y349	P29353	1.18	0.03	ACC1	S1263	Q13085_	−1.11	0.04
TRAF6	Y353	Q9Y4K3	1.18	0.03	Rab4	S199	P20338	−1.06	0.03
EGFR	T693	P00533	1.16	0.03					
PI3Kp85 B	Y605	O00459	1.16	0.04					
PAK4	S474	O96013	1.16	0.04					
p38 delta	Y182	O15264	1.15	0.01					
IRAK4	T235	P51617	1.15	0.03					
gp130	Y676	P40189	1.15	0.04					
Kit	Y568	P10721	1.15	0.05					
MEK1	S297	Q02750	1.14	0.03					
4E-BP1	S64	Q13541	1.1	0.004					
Mlk3	T277	Q16584	1.1	0.03					
PTEN	Y315	P60484	1.1	0.04					
TAK1	T187	O43318	1.03	0.04					

**Table 5 vaccines-10-00927-t005:** Differentially Phosphorylated Peptides in Node in Response to Alum.

Increased Phosphorylation					Decreased Phosphorylation				
ID	P Site	Accession	FC	*p*	ID	P Site	Accession	FC	*p*
ACC1	S29	Q13085	1.07	0.0007	SHC3	Y341	Q92529	−1.19	0.04
					NFkB-p65	S536	Q04206	−1.17	0.02
					NFAT3	S676	Q14934	−1.16	0.02
					IKK-alpha	S180	O15111	−1.15	0.02
					TAK1	S192	O43318	−1.1	0.04
					P27kip1	T1576	P46527	−1.08	0.01

**Table 6 vaccines-10-00927-t006:** Pathway Overrepresentation Analysis PCEP in Draining Lymph Node.

Pathway Name	Pathway ID	Source Name	Gene Count	*p*-Value (Corrected)
EPO signaling pathway	16151	INOH	32	1.12 × 10^33^
IL-7 signaling	16106	INOH	31	1.45 × 10^32^
JAK STAT pathway	16125	INOH	35	1.60 × 10^32^
VEGF signaling pathway	16190	INOH	29	3.32 × 10^29^
Pathways in cancer	4397	KEGG	33	3.74 × 10^27^
Signaling by Interleukins	18744	REACTOME	23	1.14 × 10^25^
TGF_beta_Receptor	15911	NETPATH	27	4.49 × 10^24^
IL2	15918	NETPATH	20	9.23 × 10^24^
BCR	15916	NETPATH	24	1.41 × 10^23^
IL6	15922	NETPATH	20	2.16 × 10^23^
Signaling by NGF	16818	REACTOME	28	3.77 × 10^23^
MAPK signaling pathway	487	KEGG	27	1.87 × 10^22^
Osteoclast differentiation	10367	KEGG	22	2.19 × 10^22^

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
