# Peer review of "Kinome Analysis to Define Mechanisms of Adjuvant Action: PCEP Induces Unique Signaling at the Injection Site and Lymph Nodes"

_vaccines, 2022, doi:10.3390/vaccines10060927_

Round 1

Reviewer 1 Report

I understand that this is a resubmission and not a revision. I see that the authors have made some textual changes in the manuscript, but my initial concerns are still there.

Major comments:

  1. It is still unclear when the kinome experiments were performed. Were these stored since 2014 or performed recently?
  2. Why is the alum group not included in the qPCR experiments?
  3. Also, the authors want to tease out the MOA of PCEP compared to other adjuvants. Yet, the kinome analysis comparisons are against the PBS-injected group. How different are the phosphorylation events when PCEP is compared to alum?

Reviewer 2 Report

Dear Editor, Thank you for sending me the manuscript for review. I think this is a a good work and may be considered for publication. My suggestion is to improve reference section including more reference from 2019-2022.

Md Abdus Subhan

SUST

Sylhet

Reviewer 3 Report

Comments to the Author

In the article titled “Kinome Analysis to Define Mechanisms of Adjuvant Action: 2 PCEP Induces Unique Signaling at the Injection Site and 3 Lymph Nodes”. The authors described the mechanisms of action and signaling response in context with the adjuvant poly[di(sodiumcarboxylatoethylphenoxy)phosphazene] (PCEP). The kinase-mediated signaling (Kinome analysis) indicates that PCEP induces a proinflammatory environment at the injection site, including activation of interferon and IL-6 signaling. They also found the elevated expression of proinflammatory genes and recruitment of myeloid cells.

Overall, the article is interesting and I am sure it will be an important addition in the area of understanding the mechanisms of action of adjuvants. I think the authors have performed all the related experiments very well. I really enjoyed reading this wonderful article. I have no questions about the manuscript, which is well-written and well-organized. The study has been well planned and designed and the methods are described properly. The results are clearly presented and discussed. There are a few minor comments that the author should consider and I would request the editor to accept this wonderful article after the revision.

Comments:

1.    Author should consider either hour or hr in the whole manuscript.

2.    Author should provide details of antibodies (like clone, catalog no. etc) used for flow analysis. This detailed information will be helpful for other researchers.

3.    I would appreciate it if the authors can provide a gating strategy for their flow cytometry analysis.

4.    Author should check typos and spelling mistakes throughout the manuscript.

5.    Author should add some relevant references from 2015 onwards. 

Round 2

Reviewer 1 Report

Responses are acceptable.

This manuscript is a resubmission of an earlier submission. The following is a list of the peer review reports and author responses from that submission.

Round 1

Reviewer 1 Report

The authors in this manuscript have done kinome analysis comparing their adjuvant with alum and PBS, the immunological data is interesting, however, the phosphorylation data are overinterpreted. Here are my comments.

Table 2 and Figure 1. It is unclear if figure 1 and table 2 are related and the data of figure 1 comes from table 2. The fold-changes values are not that great. For example, for SOC3 the fold change is 1.09 which is insignificant. How can this be considered a change from baseline etc? I think a cutoff of 1.5 fold is usually correct and ideal. Under those conditions, most of the genes didn't phosphorylate at all. 

Figure 2. Why is alum control not shown here?

Figure 3. Alum didn't show any response at all?

Table 4. In the LN there seems many genes are phosphorylated (even when cut off is considered 1.5). I think most of the action happens in LN and not in muscle. The authors should filter out the genes that have a decent fold change and see which pathways are involved.

Reviewer 2 Report

Major comments:

  1. It’s unclear if there were separate sets of mice for the RNA isolation and recruited cell experiments. The authors mention that whole muscles were either collected in TRIzol or digested for cell extraction. Please clarify and add to relevant sections.
  2. Are the phosphorylated peptides in the tables 2 ordered by p values? Please consider ordering them based on FC and then p-value for easier interpretation. Same for the tables with over-represented pathways.
  3. There are no details on sample preparation for the kinome assays. Also, since this would require specific lysis and extraction steps, was this a separate group of animals, or were the same animals, but portions of the muscle and LN, used for each of the three major assays?
  4. Were the popliteal or iliac nodes also collected? If not, why focus only on the inguinal nodes?
  5. It becomes clear only in the figure legends that the qPCR and flow cytometry experiments were performed for a 2012 and a 2014 study, wherein various timepoints were assessed after PCEP administration. Here the authors go with the 24h data from those experiments. This is not mentioned anywhere in the methods. This is very misleading and potentially problematic, as there would be significant batch effects. It is also unclear if the kinome experiments were performed on mice purchased recently or on stored materials from the 2012 and 2014 studies. If these experiments were recent, are the authors not concerned about the possible genetic drift in these animals over almost a decade since the qPCR and flow experiments were performed?
  6. If the kinome experiments were performed on new animals, why did the authors not perform the qPCR and flow experiments on these animals?
  7. As the qPCR experiments were done before the kinome analysis, they are not validations for the kinome data but rather a retrospective look at gene expression of a selected array of proinflammatory targets. Also, pathway over-representation indicates increased IL7 signaling, but IL7 is not assessed. Only a fraction of known pro-inflammatory molecules is included in the panel. The authors must include other known molecules in the analysis.

Minor comments:

  1. There are many grammar and punctuation errors and some spelling errors (e.g., line 87, splenocytes and not spleenocytes) in the text that need to be corrected.
  2. The last paragraph of the introduction should be a part of the discussion as it describes and explains core results.
  3. Figure 2, panels must be labeled, and legends provided for all three panels. Indicate which genes are statistically different.